# Isotopic evidence of acetate turnover in Precambrian continental fracture fluids

Elliott P. Mueller [1] ✉, Juliann Panehal[1], Alexander Meshoulam[1], Min Song [2], Christian T. Hansen[3], Oliver Warr [2,4], Jason Boettger[5], Verena B. Heuer [3], Wolfgang Bach [3], Kai-Uwe Hinrichs [3], John M. Eiler[1], Victoria Orphan [1], Barbara Sherwood Lollar[2,6] & Alex L. Sessions [1]

The deep continental crust represents a vast potential habitat for microbial life where its activity remains poorly constrained. Organic acids like acetate are common in these ecosystems, but their role in the subsurface carbon cycle - including the mechanism and rate of their turnover - is still unclear. Here, we develop an isotope-exchange 'clock' based on the abiotic equilibration of H-isotopes between acetate and water, which can be used to define the maximum in situ acetate residence time. We apply this technique to the fracture fluids in Birchtree and Kidd Creek mines within the Canadian Precambrian crust. At both sites, we find that acetate residence times are <1 million years and calculated a rate of turnover that could theoretically support microbial life. However, radiolytic water-rock reactions could also contribute to acetate production and degradation, a process that would have global relevance for the deep biosphere. More broadly, our study demonstrates the utility of isotope-exchange clocks in determining residence times of biomolecules with possible applications to other environments.

Fluid-bearing fractures within crystalline rocks of the Precambrian continental crust have been identified globally at sites from the Canadian Shield to the South African Craton and may store as much as one-third of the Earth's groundwater[1]. Surface meteoric water mixes with fracture fluids in the top 1–2 kilometers of the crust sustaining diverse populations of microorganisms. Here, we focus on still deeper fluids that are generally characterized by anoxia, high salinities (up to 325 g/L), low cell densities (<$10^3$–$10^5$ cells/L) and variable hydrogeologic recharge rates[2–4]. At the Kidd Creek Cu-Zn-Ag Mine (Timmins, Ontario), noble gas-derived mean residence times of fracture fluids can exceed $10^9$ years[3]. Long fluid residence times allow the products of water-rock reactions to accumulate to a greater extent than elsewhere. Despite the accumulation of these potential substrates, cell densities in the fluids are low, making the Kidd Creek Deep Fluid and Deep Life Observatory a prime window

into abiogenic synthesis[4]. Most notably, radiolysis produces abundant $H_2$ while simultaneously generating oxidants like sulfate[5–9]. At sufficiently high concentrations, $H_2$ can reduce inorganic carbon to generate methane and higher hydrocarbons through abiotic Sabatier and polymerization reactions[10–13]. It was recently suggested, based on laboratory experiments, that radiolysis in Kidd Creek may also generate simple organic acids such as acetate, formate and oxalate from water and dissolved inorganic carbon[14–16]. Indeed, the dissolved organic carbon pool in Kidd Creek's fracture waters is over 2 mM and up to 68% of this pool is composed solely of acetate and formate[16]. Through observations of Kidd Creek and other subsurface continental sites, it has become clear that abiotic water-rock reactions including radiolysis can provide a chemical framework – organic carbon, oxidants and reductants – that could support microbial communities[17].

[1]Division of Geological and Planetary Sciences, California Institute of Technology, Pasadena, CA, USA. [2]Department of Earth Sciences, University of Toronto, Toronto, ON, Canada. [3]MARUM Centre for Marine Environmental Sciences, University of Bremen, Bremen, Germany. [4]Department of Earth Sciences, University of Ottawa, Ottawa, ON, Canada. [5]Department of Earth, Environmental, and Resource Sciences, University of Texas at El Paso, El Paso, TX, USA. [6]Institut de Physique du Globe de Paris (IPGP), Université Paris Cité, 1 rue Jussieu, Paris, France. ✉e-mail: elliottpmueller@gmail.com

The synthesis mechanism of these chemical species has been studied for over thirty years at Kidd Creek, yet estimates of their turnover times are to date limited. Methane and sulfur cycling have been examined through isotopic analyses, but these measurements provide binary statements about production and consumption rather than quantitative rates[10,18]. Substrate turnover times are instead estimated via bottom-up models of radiolytic yields that come with large uncertainties[5–7,9]. Direct measurements of carbon turnover are needed for accurate evaluation of the net productivity and thus habitability of hydrogeologically isolated systems like Kidd Creek. Moreover, environmental measurements of abiogenesis rates could elucidate the quantitative importance of these reactions in other deep biosphere locations both on Earth and potentially other planets or moons.

Here, we constrain the turnover time of acetate in two deep subsurface fracture fluid systems by developing and applying an isotope-exchange clock for dissolved acetate. First, we experimentally constrained the rate of uncatalyzed (abiotic) H-isotope exchange between water and acetate methyl-H, which is presumed to occur through a tautomerization reaction[19,20]. We found that the rate of this exchange reaction follows a first-order Arrhenius relationship with temperature (Fig. 1A). Since acetate is synthesized out of H-isotopic equilibrium with surrounding fluids and exchange drives it towards equilibrium at a known rate, the apparent $^2$H-fractionation between acetate and water can serve as a clock: If acetate turnover is slower than abiotic isotopic exchange, acetate's methyl-site $\delta^2$H composition will be defined by the water $\delta^2$H and the equilibrium isotope effect (EIE) between them. Alternatively, if turnover is comparatively high, it will have a disequilibrated signature from the water. Although we do not (yet) know the magnitude of starting disequilibrium upon acetate synthesis, preventing a fully quantitative estimate of residence time, the mere presence of isotopic disequilibrium between acetate and water must indicate a residence time that is shorter than the equilibration time.

We applied this approach to fracture fluids at Kidd Creek Mine and – for comparison – at Birchtree Mine, a site with lower salinity and higher microbial activity in the Canadian Shield[16]. A suite of microbial communities with diverse metabolisms have been enriched from fluids from Thompson Mine, adjacent to the Birchtree site, including fermentation and organoclastic sulfate reduction[21]. Whereas only alkane-oxidizing and hydrogenotrophic sulfate reducers could be enriched from Kidd Creek fluids[4]. Cell densities are also higher in Thompson

fluids ($10^3$–$10^7$ cells/mL) than in Kidd Creek (<$10^4$ cells/mL)[4,21]. The distinct carbon isotope ratios of acetate in Birchtree (−27‰) and Kidd Creek (−7‰) fluids further supported the hypothesis that microbial communities were actively turning over dissolved organic molecules like acetate in Birchtree fluids, while Kidd Creek fluids represented an abiotic endmember with long organic residence times[16]. We used our isotope exchange clock method to test this hypothesis and found acetate-water $^2$H disequilibria at Birchtree that confirm acetate turn over, likely by microbial metabolisms. More notably, acetate-water disequilibria was also identified in Kidd Creek fluid, indicating relatively short acetate residence times (<1 Myr) despite fluid residence times that are 1000-times longer. Our results from Kidd Creek provide insights into an active carbon cycle within isolated deep continental fracture fluids and suggest tentative constraints on the importance of radiolytic acetate production as an abiotic reaction in the deep biosphere.

## Results and discussion

### Experimental rates of hydrogen isotope exchange between acetate methyl hydrogen and water

Acetate was incubated at temperatures between 60 °C and 200 °C in the presence of 5% deuterated water in pressurized gold bags (see Methods). To derive the kinetic rate constant for hydrogen exchange between acetate's methyl group and ambient water, the $^2$H/$^1$H ratio ($\delta^2$H value) of acetate's methyl group was measured periodically throughout the incubations via ESI-Orbitrap mass spectrometry (See Methods)[22]. Under every condition tested, acetate $\delta^2$H values increased with time reflecting exchange with the $^2$H-enriched aqueous medium. At high temperatures (≥150 °C), the rate of acetate $^2$H enrichment over time was initially linear then gradually flattened as it approached isotopic equilibrium with water (Fig. S2). At lower temperatures, the exchange kinetics were too slow to allow full equilibration of acetate and water within the runtime of the experiments. The fitted half-times for exchange increased exponentially with decreasing temperature from 3 hours to 810 years, following an Arrhenius relationship ($R^2$ = 0.999, $E_A$ = 138 kJ/mol, Fig. 1). Replicate incubations, which were performed for all conditions except 100 °C, resulted in similar reaction rates (overlapping data points in Fig. 1, Table S3). Exchange between acetate's methyl-site and water is presumed to occur through a reversible tautomerization between ethanoate and ethenol moieties (Fig. S4). Regardless of the exact

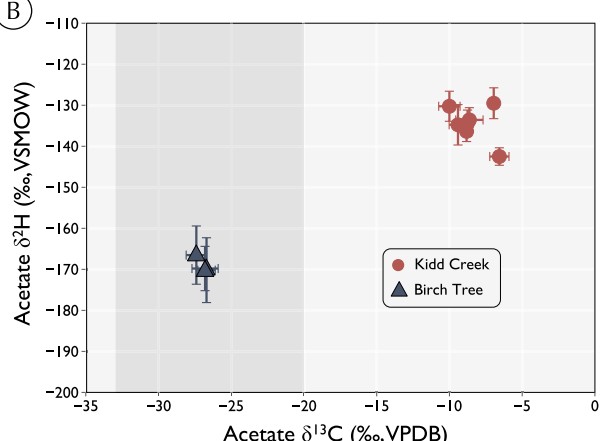

**Fig. 1 | Acetate exchanges hydrogen isotopes with water at a temperature-dependent rate. A** Arrhenius plot of hydrogen isotope exchange rates with a linear regression through experiments at 60 °C (*n* = 3), 100 °C (*n* = 1), 150 °C (*n* = 2) and 200 °C (*n* = 2) (solid circles). Extrapolated reaction rates are projected to 25 °C (open circle). Shaded region represents 2 RMSD. **B** Carbon and hydrogen isotope composition of acetate from Kidd Creek and Birchtree mines. Shaded regions represent $\delta^{13}$C of total organic carbon from the metasedimentary rocks of the Kidd Creek formation[34]. Error bars reflect standard deviation on analytical triplicates.

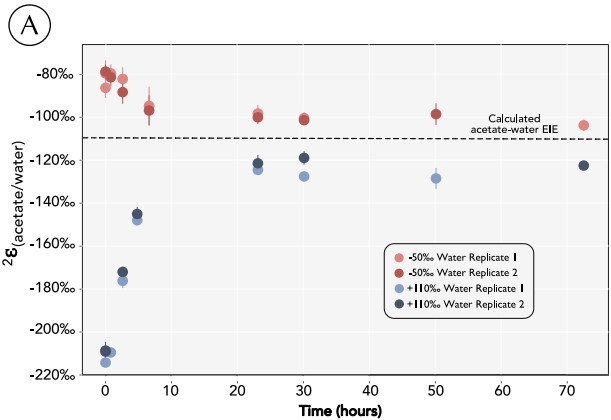

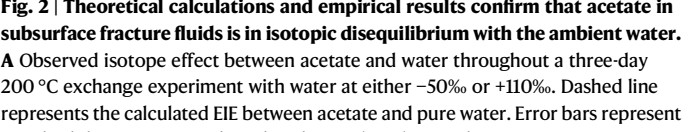

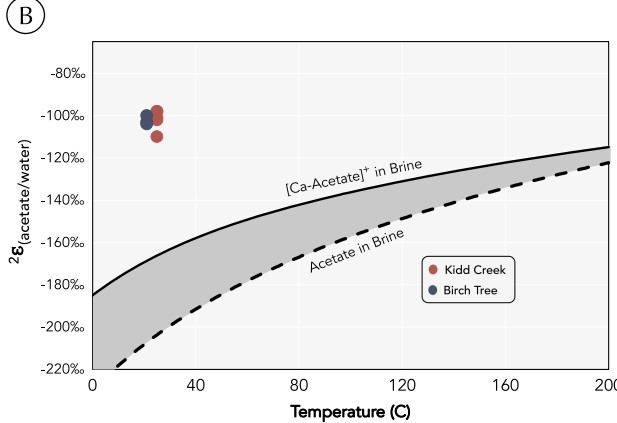

**Fig. 2 | Theoretical calculations and empirical results confirm that acetate in subsurface fracture fluids is in isotopic disequilibrium with the ambient water.** **A** Observed isotope effect between acetate and water throughout a three-day 200 °C exchange experiment with water at either −50‰ or +110‰. Dashed line represents the calculated EIE between acetate and pure water. Error bars represent standard deviation on analytical replicates ($n = 3$). **B** Hydrogen isotope

fractionation between acetate and water ($^2\varepsilon_{acetate/water}$) at both sites. Solid line is the calculated EIE between the Ca-acetate complex and brine water. Dashed line is the EIE between free acetate and brine water as a function of temperature. Error bars are covered by the data points and represent standard deviation on analytical replicates ($n = 3$).

mechanism, the excellent fit to an Arrhenius relationship between 60 °C and 200 °C suggests that the mechanism of exchange does not change within the tested temperature range. Extrapolating to the ambient temperature for samples collected at Kidd Creek and Birchtree (25 °C), the predicted exchange half-time was $250,000 \pm 70,000$ years (2xRMSD).

## Determining the equilibrium isotope effect
Equilibrium $^2$H-isotope effects (EIEs) for acetate-water were calculated using density functional theory (DFT) across a range of temperatures (see Methods). These indicated a temperature-dependent change in the EIE from −108‰ at 250 °C to −192‰ at 25 °C (Fig. 2B). Four high temperature incubations at 200 °C were designed to experimentally test these calculations. Incubations were started with varying magnitudes and direction of isotopic disequilibrium, but in each case acetate $\delta^2$H values changed with time until the experiments converged to similar EIEs. Water was present in excess and so did not change in $\delta^2$H value. Equilibrium was reached in less than one day at 200 °C and remained there for two days (Fig. 2A). On average, the measured EIE ($0.888 \pm 0.012$) was within analytical error of the DFT-calculated value (0.882). While the two experimental series did not perfectly converge in $\delta^2$H values, they came within ~20‰ of each other. This offset is potentially due to analytical artifacts associated with measuring the high $\delta^2$H value of acetate in the $^2$H$_2$O spiked sample and is small in comparison to the scale of natural hydrogen isotope variations (blue, Fig. 2A). Thus, at 200 °C, the empirically determined EIE corroborates the DFT calculations.

## Carbon and hydrogen isotope compositions of acetate from deep mines
The $\delta^{13}$C and $\delta^2$H values of acetate extracted from Kidd Creek and Birchtree fracture fluids were measured via the ESI-Orbitrap method, revealing different isotopic compositions at the two sites[22]. Samples collected from three separate boreholes in Kidd Creek between 2008 and 2018 yielded $\delta^{13}$C values of −10.0‰ to −6.6‰ (VPDB) and $\delta^2$H values of −142‰ to −130‰ (VSMOW). In contrast, acetate extracted from three fracture fluid samples from Birchtree yielded $\delta^{13}$C values of -26.7‰ to −27.4‰ and $\delta^2$H values of −167‰ to −170‰ (Fig. 2 and Table S2). All $\delta^{13}$C values match the range of values previously reported for these two sites[16]. When compared to the previously-measured $\delta^2$H values of water from Kidd Creek and Birchtree (−36‰ and −74‰, respectively)[19], a similar apparent hydrogen isotope fractionation

between acetate and water exists at both sites. This fractionation ranges from −115‰ to −90‰ (Fig. 2B) and differs from isotopic equilibrium at 25 °C by over 50‰. These data demonstrate that acetate in Kidd Creek and Birchtree fracture fluids is far from the calculated H-isotopic equilibrium with water and must therefore have rates of production and consumption that are faster than the rate of abiotic exchange.

The identical apparent acetate-water hydrogen isotope effect ($^2\varepsilon_{acetate/water}$) from the two sites is notable (Fig. 2B). One possibility that we considered is whether complexation of acetate by the abundant (>1 M) dissolved cations[4] could significantly alter the EIE, i.e. a 'matrix effect'. In this case, a shared $^2\varepsilon_{acetate/water}$ value between the sites would be possible if acetate at both sites was in equilibrium with water and the $^2\varepsilon_{acetate/water}$ value matched the shifted EIE. Calcium is the most abundant cation in Kidd Creek and Birchtree fluids that complexes with free acetate, thus the Ca-acetate complex represents the most likely acetate complexation in these systems. To test whether complexation shifts the calculated EIE, we calculated the partition function ratio of a calcium-acetate bidentate complex and for high ionic strength brines then combined these to define an EIE for the complex-brine equilibrium. Conservatively assuming that all the acetate is ligated to calcium cations and is in equilibrium with a CaCl$_2$ brine, the calculated EIE is −167‰ at 25 °C, which is 60‰ offset from the fractionation observed in Kidd Creek and Birchtree (Fig. 2B). Thus, a comparison of DFT calculations and environmental data suggest that acetate and water in Kidd Creek and Birchtree are in substantial isotopic disequilibrium, whether acetate exists as a free anion or is complexed to calcium in solution. The identical value of $^2\varepsilon_{acetate/water}$ values observed at both sites (Fig. 2B) may instead reflect kinetic isotope effects that provide insight into acetate turnover mechanisms.

## Acetate is cycled in the continental deep subsurface
The turnover times of organic molecules can provide important constraints on the productivity and habitability of isolated systems like the continental deep biosphere, but to date such timescales have been difficult to measure[17]. Water-rock reactions influencing the geochemistry of Kidd Creek and other sites often operate too slowly to replicate through experimentation. Similarly, microbial growth rates and metabolic fluxes typical of these settings are inaccessibly slow on laboratory timescales[2]. While these processes can be identified through isotope geochemistry and genomic analyses, rates of abiogenesis and/or microbial metabolism remain elusive[23]. Our new

H-isotope exchange clock helps to fill that gap by setting upper limits on residence times (i.e. lower limits on production and consumption rates) for acetate. Moreover, the general approach should be directly applicable to other organic molecules in the environment

including many potentially important organic substrates and biomolecules.

In fracture fluids from both Kidd Creek and Birchtree, isotopic disequilibrium between acetate and water implies active production and consumption of acetate by physical, chemical, and/or biological processes. These processes must generate and consume acetate faster than the abiotic exchange reaction can establish H-isotope equilibrium with water. Given that equilibration of hydrogen atoms occurs in less than four half-times, acetate residence times must be less than one million years, at least 1000-fold shorter than that of Kidd Creek fracture fluids (>1 Gyr). Normalizing by the concentrations of acetate (Table S2) and assuming present-day concentrations are at steady-state, these turnover times require acetate production and consumption rates of >1 nM/year and >0.1 nM/year in Kidd Creek and Birchtree, respectively. Since estimated physical fluid recharge rates are slower than acetate turnover times[3], our data suggest active production and consumption of acetate by microbial metabolisms and/or abiotic reactions.

## Acetate consumption could support microbial communities

Many anaerobic microorganisms use acetate as a carbon and electron source. The rates of acetate consumption implied by our residence time estimates provide an opportunity to quantify the amount of metabolic power potentially available to microbes consuming this substrate in the continental deep biosphere. Anaerobic respiration – represented here as sulfate reduction – and methanogenesis are common acetate consumption pathways in anoxic environments[24,25]. Considering the lower threshold of 1 nM/year for acetate consumption in Kidd Creek, acetate would supply $10^{-11.5}$ W/L or $10^{-12}$ W/L via sulfate reduction or methanogenesis, respectively (Fig. 3). Assuming a range of cell-specific maintenance powers (the flux of energy required to maintain a cell)[26–28], this rate could support between $10^2$ to $10^6$ cells/mL (Fig. 3). In saline fracture fluids of the continental subsurface, microbial cells must synthesize organic osmolytes to combat high osmotic pressures, increasing their basal power demands[29,30]. Our results suggest that even with these higher power requirements, at least $10^3$ cells/mL could theoretically survive solely on acetotrophic metabolic pathways in Kidd Creek (Fig. 3). However, such calculations only reveal the viability of these prospective metabolic pathways and cannot be used as sole evidence of microbial acetotrophy. Further evidence is required to determine whether acetate is actively being consumed by biotic processes.

## Constraining acetate sources and sinks in the subsurface

The processes producing and degrading acetate can be constrained using its steady-state isotopic composition. In anoxic settings, acetate typically has a $\delta^{13}C$ value similar to that of the surrounding total organic carbon (TOC). This is commonly attributed to minimal isotope effects associated with the production of acetate by microbial fermentation and consumption by anaerobic respiration[31–33]. Acetate in Birchtree fracture fluids has $\delta^{13}C$ values that match this expectation, but it does not have the characteristic $^{13}C$ and $^2H$ depletion associated with chemolithoautotrophic acetogenesis[22,33–35]. This suggests that acetate turnover in Birchtree fluids is driven by heterotrophic microbial metabolisms.

In contrast, acetate in Kidd Creek is $^{13}C$-enriched relative to TOC[36]. If microbial activity is similarly responsible for acetate turnover in Kidd Creek fracture fluids, the reactions(s) consuming acetate must have larger (normal) carbon isotope effects than those in Birchtree. Acetoclastic methanogenesis exhibits such an isotope effect (25-30‰)[37]. When the fermentation of organic matter to acetate is coupled with methanogenic consumption, acetate can indeed be $^{13}C$-enriched relative to TOC; however, this enrichment is not consistent across environments and the mechanisms behind it are still unclear[25,32,38,39]. Furthermore, the isotopic composition of methane and low ratio of methane-to-higher-alkanes in Kidd Creek fluids are not consistent with the significant rates of acetoclastic methanogenesis required to generate the observed $^{13}C$ enrichment in acetate[11,37]. During cultivation studies, autotrophic and alkane-oxidizing sulfate reducers were enriched from Kidd Creek fluids, but fermentative and acetoclastic methanogenic microorganisms were not[4]. Importantly, the lack of microbial growth does not preclude these metabolic niches from being an important component of the ecosystem. When culture-independent 16S rRNA sequencing was performed on the same borehole fluids, a variety of putatively chemolithoautotrophic and organisms were identified, including *Fuchsiella ferrireducens*, an iron-reducing bacterium capable of reductive acetogenesis[40]. While acetogenesis is a possible source of acetate in these systems, cultured acetogens consistently generate $^{13}C$ and $^2H$ depleted acetate, the opposite signal to what is observed here in Kidd Creek fluids[22,31–35].

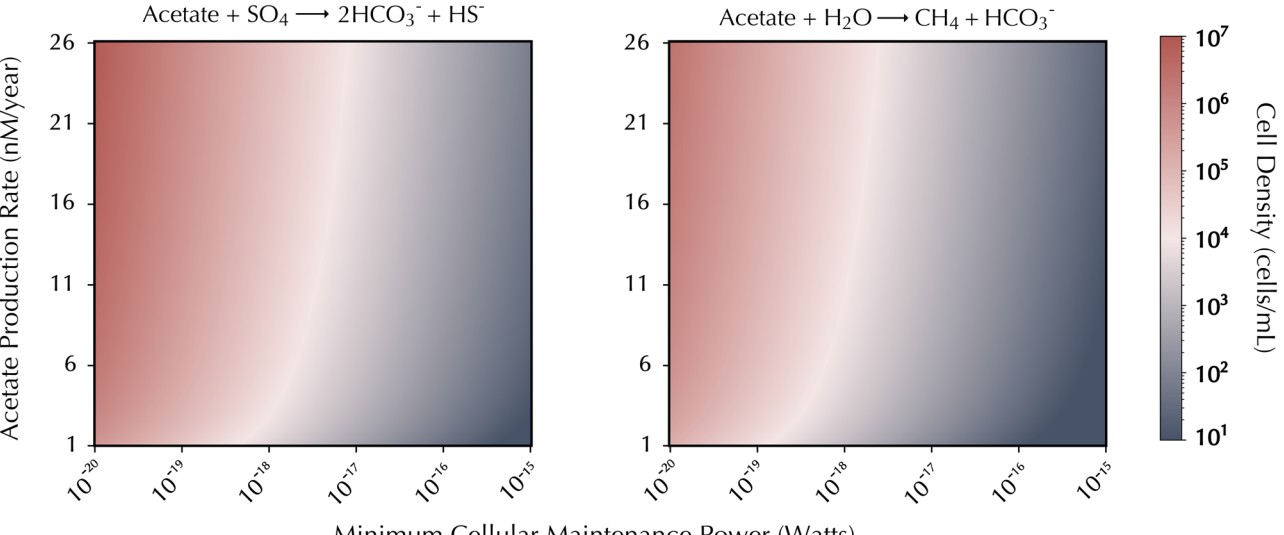

**Fig. 3 | Acetate cycling could theoretically support microbial communities in the continental subsurface.** Theoretical cell densities for sulfate reducers (left) and acetoclastic methanogens (right) that could be supported in the fracture fluids over a range of acetate production rates.

As such, other mechanisms should be considered to explain acetate turnover in this system.

The identical hydrogen isotope fractionations between acetate and water at Kidd Creek and Birchtree could indicate turnover mechanisms that are shared between the mines, such as radiolytic reactions. Radiolysis is well documented in the deep biosphere and has been shown to both produce and degrade acetate in laboratory experiments[14,15]. Radiolytic reactions occur when alpha, beta and gamma irradiation from natural decay of U, Th and K in the rock matrix triggers reactions with surrounding water, solutes, and minerals[8,35]. Since radiolysis drives substantial abiotic chemistry in subsurface fluids (i.e. $H_2$ production[5,7]), it could produce acetate in situ as well[14–16].

If radiolytic synthesis is the source of acetate in these fracture fluids, it operates at a rate that far exceeds those observed in laboratory studies. Maximum net yield during in vitro experiments is 6 nM acetate per joule of alpha radiation, corresponding to 0.007 nM/yr acetate generation rate in Kidd Creek fluids (see Methods), well below the minimum production rate estimated here[15]. These results should be interpreted with caution though. Radiolytic synthesis of organic acids is not a single production reaction but a network of reactions that both creates and degrades acetate[14,15]. The net yield measured in vitro represents a balance of production and degradation fluxes, whereas gross yields could be much higher. Thus, if radiolysis is both producing and degrading acetate in situ, it could support fast turnover times without having high net generation rates. Kinetic isotope effects associated with this turnover could then explain the constant hydrogen isotope fractionation from water observed at both sites. However, while radiolysis is likely cycling acetate in the continental subsurface to some extent, we cannot presently determine whether it is *solely* responsible for acetate turnover based on our current understanding.

Future work should carefully examine radiolytic reactions under conditions that match the subsurface to assess their rates of acetate turnover and associated isotope effects. Given that the substrates for radiolysis – water and DIC – are ubiquitous, this process could provide a means to fuel acetotrophic metabolisms in environments well beyond the Precambrian continental subsurface, including global marine sediments, groundwaters, and the subsurface of other planets or moons.

### Isotope-exchange clocks may have wide-ranging applications

Isotope-exchange clocks may also be relevant for other molecules and environments. For isolated systems characterized by slow turnover (i.e. subsurface environments of Earth, Mars or Europa), the acetate H-exchange reaction introduced here could be a useful constraint on acetate residence times or could simply confirm the presence of an active carbon cycle. However, for more biologically productive environments with fast turnover of organics (i.e. shallow marine sediments[41]), this particular clock is insensitive. Isotope exchange in organic molecules that experience more rapid equilibration of C-bound H would provide more useful information about substrate turnover in these systems. Molecules containing acidic alpha-H atoms, which can undergo tautomerization more easily than acetate (i.e. longer chain organic acids and aldehydes), are potential targets[42]. Conversely, molecules with yet slower exchange (e.g. alkanes) could provide information about turnover in hotter environments[43]. Our study provides the analytical and experimental basis for developing these techniques and directly constraining the turnover of small biomolecules in situ using their hydrogen isotope composition, one that could be applied to diverse environments.

## Methods

### Organic acid extraction

Organic acids were extracted following the procedure developed by Mueller et al. [22] with minor changes to account for the high concentrations of chloride in the fracture waters. Briefly, samples of fracture fluid were titrated to pH >6 with NaOH if necessary. Samples were run through a Dionex Ag/H cartridge at 0.5 mL/min to remove chloride after the cartridge had been washed with 300 mL purified water (MilliQ) at 2 mL/min. The first 0.5 mL of eluent from the cartridge was discarded as it represented the dead volume. The remaining sample was collected until almost all the resin was used, carefully avoiding over-filling the cartridge, which would cause chloride to leak through. The cartridge eluent was injected onto a Dionex high performance ion chromatography instrument with an AG-11HC column and a KOH gradient from 1 to 20 mM. The organic acid fraction of the chromatogram was collected into vials using manual fraction collection. This step was repeated for samples with lower acetate concentration and collected into the same vial. The collected acids were titrated to pH >6 with degassed, anoxic NaOH and then dried down under nitrogen. Samples were redissolved in LC-MS grade methanol.

### Stable isotope analysis

The majority of samples were analyzed on a heated electrospray ionization (HESI) Orbitrap QExactive HF (Thermo Fisher, Bremen, Germany) following the protocol of Mueller et al. [22]. This technique quantifies the molecular-average $\delta^{13}C$ (VPDB) and methyl-specific $\delta^2H$ of acetate by comparison to an working standard of sodium acetate ($\delta^{13}C = -19.2‰$, $\delta^2H = -127‰$). Certain samples were measured on an electrospray ionization (ESI) Orbitrap Exploris 240, but the mass spectrometry parameters were identical and the same standard was used for all measurements. Multiple sample introduction methods into the Orbitrap were used throughout the course of this study.

For direct infusion measurements, 500 µL syringe (Hamilton) was filled with sample or standard solution (in LC-MS grade methanol) and attached to a syringe pump (Chemyx). Solution was infused into the mass spectrometer at 5 µL/min. After a 7-minute acquisition, the syringe and its tubing were washed with 2 mL of LC-MS grade methanol and the next sample or standard was loaded into the syringe pump. This was repeated to achieve bracketed, sample-standard comparisons (AAAABBBBAAAA, A = standard replicates, B = sample replicates). This method was used when memory effects between sample and standard due to large differences in $\delta^2H$ or $\delta^{13}C$ were a concern. This was especially important for $^2H$-enriched acetate samples from exchange experiments.

For dual inlet measurements, two 500 µL syringes (Hamilton) were filled, one with sample and the other with standard solution (in LC-MS grade methanol) and attached to a syringe pump (Chemyx). The solution was infused into the mass spectrometer at 5 µL/min. Using a Rheodyne 6-port valve, sample and standard were alternated while achieving continuous flow of both (after Hilkert et al.)[44]. Each acquisition block was 12 minutes with 4-5 minute switch times between blocks cut out of the data acquisition to avoid carryover effects. This was repeated to achieve bracketed, sample-standard comparisons (ABABABA, A = standard replicates, B = sample replicates). This method was used for the majority of Kidd Creek and Birchtree samples. Acetate standard was diluted to match sample ion current.

For in-flow injection measurements, samples were infused into the mass spectrometer using a Vanquish Horizons HPLC Split Sampler Autosampler and a Vanquish Horizons Pump set to 5 µL/min with degassed LC-MS grade methanol as an eluent. An injection volume of 50 µL was used to insert this sample into the flow of methanol which carried it to the Orbitrap for 14 min. At that time the flow rate was increased to 30 µL/min to clear residual sample from the transfer lines. At 18.5 minutes, the flow rate was dropped again to 5 µl/min and after 90 s, the next injection began. Data acquisition included all 20 min of the run but only integrated between 2 and 12 min to calculate isotope ratios. This was repeated to achieve bracketed, sample-standard comparisons (ABABABA, A = standard

replicates, B = sample replicates). Acetate standard was diluted to match sample ion current.

In all of the above methods, the following ESI parameters were used as default. Minor adjustments were made daily to tune the instrument for spray stability. Polarity = negative, spray voltage = 3.0 kV, spray current <0.2 μA, Auxiliary gas = 1 (arbitrary units), sweep gas = 1 (arbitrary units), sheath gas = 10 (arbitrary units), auxiliary gas temperature = 100 °C, RF lens = 60%, capillary temperature = 320 °C. The following Orbitrap parameters were used for all analyses. Automated gain control = 1e6, resolution = 60,000, microscans = 1, quadrupole range = 57–62 m/z, lock mass = off. Raw data off the Orbitrap was extracted using the software IsoX (Thermo Fisher, Bremen, Germany) and converted to isotope ratios using a Python script. This script uses the Makarov equation outlined in Mueller et al. [22] to convert from ion intensities to ion counts. It then culls scans that are >99th percentile or <1st percentile in total ion current to avoid integrating scans with ion source aberrations.

### Exchange reactions

High-temperature acetate-water exchange experiments were conducted using a customized Dickson-type flexible reaction cell setup (Parr Instruments) with no vapor phase present. Each flexible gold bag was filled with 90 mL of 1 mM sodium acetate in MilliQ water (pH 6-7) that was sparged with nitrogen and pressurized to 30 MPa. Two experiments were performed at 150 °C in 5% $^2H_2O$. One was run for a week, sampling every 24 h, while the other was run for a month, sampling every 3–5 days. Another month-long experiment with 5% $^2H_2O$ was performed at 100 °C, sampling every 3–5 days. Acetate-water exchange experiments were also performed at 60 °C in 60 mL serum vials. Each vial was filled with 50 mL of 1 mM sodium acetate in 5% $2H_2O$ (pH 7) that had been sparged with nitrogen and sealed with a butyl rubber stopper and crimped with an aluminum cap. At each timepoint, 1 mL of sample was collected via needle and syringe. The sample was immediately frozen and stored at −20 °C and the solution was sparged with nitrogen again to remove any air introduced during sampling. These experiments were done in triplicate. All exchange experiments were performed at pH 6–7 to match environmental conditions.

Additional high temperature flexible gold bag experiments were performed to determine the equilibrium isotope effect at 200 °C (30 MPa). Each reaction cell was filled with 90 mL of 1 mM sodium acetate (pH 6-7) in either −50‰ or +110‰ $δ^2H$ water. Each condition was measured in duplicate, resulting in four total experiments. Samples were taken every hour for the first six hours to measure the extent of isotopic exchange with time and then every ~6–12 h for the next 66 h. At each time point, 1.5 mL of the sample was collected and discarded to remove the dead volume from the sampling apparatus and then an additional 1.5 mL of sample was taken for acetate $δ^{13}C$ and $δ^2H$ analyses. Collected aliquots were immediately frozen and stored at −20 °C until they were analyzed.

The kinetic rate constants for H-isotope exchange were calculated using the formulation from Sessions et al. [43]:

$$\frac{F_e - F_t}{F_e - F_i} = e^{-kt} \tag{1}$$

where $F_t$ is the $^2H$ fractional abundance (i.e., mole fraction) at a given timepoint, $F_i$ is the initial fractional abundance and $F_e$ is the fractional abundance at equilibrium. The latter was calculated using the fractional abundance of the water and the equilibrium isotope effect from DFT models at the corresponding temperature. In experiments where the isotope composition approaches or reaches equilibrium, data points close to the equilibrium value were discarded from the calculation of rate constant due to the large propagated errors when the natural logarithm of the value $F_e - F_t$ was close to zero.

### Isotope fractionation calculations

The apparent hydrogen isotope fractionation between acetate and water ($ε_{acetate/water}$) was calculated as:

$$^2\alpha_{acetate/water} = \frac{\delta^2H_{Acetate} + 1000}{\delta^2H_{Water} + 1000} \tag{2}$$

$$^2\varepsilon_{acetate/water} = \left( ^2\alpha_{acetate/water} - 1 \right) \times 1000 \tag{3}$$

### Thermodynamics and cell density calculations

The free energy (ΔG) available to microbial metabolisms was calculated by adjusting the standard free energy (ΔG°) for the activity of the reactants and products found in Kidd Creek fracture fluids following the equation:

$$\Delta G = \Delta G° + RT\ln Q \tag{4}$$

where R is the ideal gas constant (kJ/mol/K) and T is temperature (K), set to 298 K at 500 bar pressure. Q is the reaction quotient defined as:

$$Q = \prod a_{a_i}^{v_i} \tag{5}$$

where a is the activity of a substrate defined as the product of its concentration (molar) and gamma value and v is the stoichiometric coefficient which is negative for reactants. Gamma values for sulfate, methane and bicarbonate were found on the Geochemists Workbench with the thermo-hmw.dat database, which uses a Pitzer equation based Harvie-Møller-Weare activity model owing to the high ionic strength of the fracture fluid (4.9 molal). Acetate is not part of this database, so it was calculated with extended Debye Hueckel equation using the thermo.dat database. The concentrations used in these calculations were taken from data in Lin et al.[10]. Sulfate, bicarbonate, acetate and methane concentrations were set to 620 μM, 57 μM, 1.3 mM and 2.1 mM, respectively. Methane concentration was calculated from fluid flow rate, gas exsolution rate from the fluid, and the concentration of methane in the gas (from Lin et al.[10]). It was assumed that all methane was dissolved fully in solution due to the high (500 bar) in situ pressure of the fracture fluids (after Sherwood-Lollar et al. [11]). Sulfide was below detection limits (<2 μM). Its concentration was set to 10 nM but increasing its concentration to the detection limit did not change the implications of the cell densities (>10 cells/mL at all maintenance energies simulated).

Cell density (cells/L) is calculated by combining the acetate turnover rate (M/s), the free energy of the reaction (J/mol), and the maintenance energy of a cell (J/s/cell).

$$\rho = \frac{\tau_{AC} \times \Delta G}{ME} \tag{6}$$

where $τ_{AC}$ is the turnover time and $ρ$ is the cell density.

### Density functional theory calculations of EIE

Temperature-dependent $^2H/^1H$ equilibrium fractionation between acetate and water was estimated using density functional theory. Liquid-phase acetate and water molecular models were optimized in the GAUSSIAN(TM) program, revision D.01 and GAUSSIAN 16, revision B.01 using basis set 6-311 G(d,p)[45,46] and functional B3LYP under Tight optimization criteria (maximum/RMS atomic displacement 0.00006/0.00004 Bohr, maximum/RMS force 0.000015/0.00001 Hartrees/Bohr or Hartrees/Radian), with an Ultrafine integration grid mesh. The integral equation-formalism polarizable continuum model was used to represent the solvation environment[47,48]. Following

optimization, frequency calculations were carried out for the monoisotopic isotopologues and with a single $^2H/^1H$ substitution to determine the effect of $^2H/^1H$ substitution on vibrational frequencies. The Urey-Bigeleisen-Mayer equation was used to calculate the temperature-dependent reduced partition function ratio of each species under $^2H/^1H$ substitution[49]. Corrected ratios were computed using the temperature-dependent regression of Wang et al.[48] to account for the effects of anharmonicity[50]. The equilibrium fractionation factor was then computed as the ratio of the corrected ratios at the desired temperature.

Temperature-dependent $^2H/^1H$ equilibrium fractionation between the Ca-acetate complex and water was estimated using an empirically derived molecular geometry for the complex, which was then optimized using the same level of theory and basis sets as in the DFT calculations above[51]. More details regarding these calculations can be found in the Supplemental Information (Tables S4-S6). The partition function ratio of water was adjusted to account for the 'salt effect' of a 3 M $CaCl_2$ brine, which was empirically determined to be 15‰ at 25 C by Horita et al.[52]. The beta factor for water ($\beta_{water}$) was multiplied by 1.015 to ascertain the beta factor of the brine ($\beta_{brine}$):

$$\beta_{brine} = 1.015 \times \beta_{water} \qquad (7)$$

The beta factors for free acetate ($\beta_{acetate}$) and Ca-acetate complex ($\beta_{Ca-acetate}$), calculated from the DFT simulations, were then used to calculate the EIE between acetate and water ($\alpha_{acetate/water}$), between acetate and brine ($\alpha_{acetate/brine}$) and between the Ca-acetate complex and brine ($\alpha_{Ca-acetate/brine}$). For example, the EIE of Ca-acetate complex and brine is calculated as such:

$$^2\alpha_{complex/brine} = \frac{\beta_{complex}}{\beta_{brine}} \qquad (8)$$

$$^2\varepsilon_{complex/brine} = \left( ^2\alpha_{complex/brine} - 1 \right) * 1000 \qquad (9)$$

**Radiolytic yield calculations**
To estimate the radiolytic yield (nM/J) of acetate production by alpha, gamma and beta irradiation in Kidd Creek needed to support a given rate of acetate production, modified calculations from Warr et al.[9] were used. The total acetate yield ($Y_{AC}$) in nM/s is defined as:

$$Y_{AC} = \frac{\sum E_{net,i} \times G_i \times \rho_{bulk}}{\phi} \qquad (10)$$

where i represents either alpha, gamma or beta radiation and $E_{net}$ is the dose rate (Gy/s) and G is the radiolytic yield (G). The bulk rock density ($\rho_{bulk}$) was set to 2.98 kg/dm³. $\phi$ is the porosity, typically ~1% at crystalline rocks sites like Kidd Creek[9]. Here, we assume that beta and gamma radiation does not produce acetate, since it has not been measured, such studies have not yet been done, so only α radiation is considered. Consequently, this represents a conservative estimate of radiolytic acetate production. Alpha radiolytic yields were taken from Vandenborre et al.[15]. In experiments with 200 μM dissolved carbonate in pure water, acetate accumulated to 8 μM within 1400 Gy of absorbed radiation and plateaued at this concentration up to 5600 Gy, due to competing production and consumption reactions reaching a steady state. This results in a range of 1.3 to 6.0 nM/J for alpha radiation yields.

The dosage rate of alpha radiation is calculated as:

$$E_{net,\alpha} = \sum \frac{E_{\alpha,X} \times W \times S_{\alpha}}{1 + W \times S_{\alpha}} \qquad (11)$$

Where $E_\alpha$ is the dosage of alpha radiation emitted (Gy/s) and $X$ represents the specific elemental source of that radiation. $S_\alpha$ is the stopping power of rock to alpha radiation set at 1.5 after Warr et al.[9]. $W$ is the water-rock ratio set to 0.37% calculated following Warr et al.[9], using water and rock density of 1.11 g/cm³ and 2.98 g/cm³, respectively, and a porosity value of 1%[5,9].

At 1% K, 1 ppm Th and 1 ppm U, these elements emit 0, $1.93 \times 10^{-12}$ and $6.9 \times 10^{-12}$ Gy/s of alpha radiation, respectively[9]. To estimate $E_\alpha$ for each of these elements in Kidd Creek, they were linearly increased based on the actual concentration in the deposit, which are 1.5 ppm, 6.7 ppm and 1.7% for U, Th and K, respectively[9]. Therefore, the $E_\alpha$ for U, Th and K is estimated at 0, $1.3 \times 10^{-11}$ and $1.0 \times 10^{-11}$ Gy/s in Kidd Creek.

**Reporting summary**
Further information on research design is available in the Nature Portfolio Reporting Summary linked to this article.

## Data availability
The data files generated during ESI-Orbitrap analysis of the Kidd Creek and Birchtree fracture fluids is provided in a public GitHub repository (https://doi.org/10.5281/zenodo.13798759)[53]. Any additional data beyond those found in the repository can be made available on request.

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

## Acknowledgements

We would like to thank Nathan Dalleska (Caltech) for helpful discussions about sample processing as well as Andreas Hilkert and Dieter Juchelka (Thermo Fisher, Bremen) and the Caltech Proteome Exploration Lab for use of their Orbitrap facilities. Funding for this work came from an NSF Gradaute Research Fellowship DGE-1745301 (to E.P.M.), a European Association of Organic Geochemistry Research Award (to E.P.M.), the NASA Astrobiology Institute grant # 80NSSC18M0094 (to J.M.E and A.L.S.), a CIFAR Earth 4D grant (to B.S.L. and V.O). This work was also supported by the Deutsche Forschungsgemeinschaft through the Cluster of Excellence "The Ocean Floor – Earth's Uncharted Interface" (project 390741603) to V.H.

## Author contributions

E.P.M. conceptualized and designed the study and performed data analysis. E.P.M., J.P. and M.S. performed sample chemical preparation and Orbitrap analysis. E.P.M., C.H. and V.H. performed isotope exchange reactions. J.B. and A.M. performed DFT calculations. J.E., A.L.S, V.O, B.S.L, K.H., O.W. and W.B. provided laboratory analytical facilities and samples as well as important scientific insights. All authors contributed to data interpretation and manuscript writing.

## Competing interests

The authors claim no competing interests.
