## [Transparent Peer Review file · Nature Communications]

Isotopic evidence of acetate turnover in Precambrian continental fracture fluids

Corresponding Author: Mr Elliott Mueller

Version 0:

Reviewer comments:

Reviewer #1

(Remarks to the Author)

The manuscript by Mueller et al reports the development of a hydrogen isotope exchange clock and its application to constrain the turnover time of acetate in Precambrian fracture fluids. The manuscript is clear and well written, and the subject matter and potential impact are suitable for Nature Communications. The development of the isotope exchange clock is supported by a strong experimental dataset and, as the authors suggest, this novel approach can be generalized across a range of compounds and timescales. The work therefore has potential to be an impactful contribution. However, in my opinion, the paper's primary conclusion concerning acetate turnover in Precambrian crustal fluids is not adequately supported. Specifically, as detailed below, that conclusion depends on establishing that acetate and water in the crustal fluids are in substantial hydrogen isotopic disequilibrium. The presently available data and accompanying discussion do not convincingly do so.

Isotope exchange clock:

The manuscript describes a series of time course deuterium labelling experiments that beautifully demonstrate exchange of protons between the methyl group of acetate and the water matrix, ultimately leading to isotopic equilibration between the two pools. The experiments further demonstrate a temperature dependence in the rate of isotopic exchange that adheres to a first order Arrhenius relationship over a temperature range from 60-200C. This relationship is used to place an upper limit on the rate of isotopic exchange at 25C. This is a novel approach that could have wide application in natural systems because, as the authors note, it is generalizable to a range of different compounds and equilibration timescales. This is an important and potentially impactful contribution.

Acetate turnover rates in natural systems:

The newly developed isotope exchange clock is used to place upper limits on the turnover time of acetate in fracture fluids from Kidd Creek and Birchtree mines, which differ in terms of microbial activity and the concentrations and carbon and hydrogen stable isotopic composition of acetate. It is argued that acetate is out of isotopic equilibrium with respect to the host fluids in both systems, such that the acetate turnover time must be shorter than the isotope equilibration timescale (< 1Myr). This is the major conclusion of the paper, and the subsequent discussion is focused on exploring the possible mechanisms and implications of acetate turnover. This conclusion depends on demonstrating that a hydrogen isotopic disequilibrium exists between acetate and water. In my opinion, this is not convincingly demonstrated. The expected equilibrium isotope effect (EIE) at the mine fluid temperature of 25C is calculated via DFT, which was also used to calculate the EIE at temperatures ranging up to 200C. Experiments conducted at 200C demonstrate that acetate and MilliQ water equilibrate with an EIE that is within analytical error of the DFT-calculated value. This stands as a solid experimental demonstration that DFT accurately predicts the EIE at 200C in a pure water matrix. To demonstrate acetate-water disequilibrium in the mine fluids requires that the DFT calculations also accurately reflect the EIE at 25C in high ionic strength fluids.

The authors note that Kidd Creek and Birchtree exhibit nearly identical acetate-water hydrogen isotope fractionation, despite the considerable differences that led to their being chosen as contrasting sites for the present study. Two explanations are considered. The favored explanation is that the identical fractionation "likely reflects kinetic isotope effects that provide insight into acetate turnover mechanisms". This certainly cannot be ruled out, but it would require that such effects yield identical fractionation in sites that differ in the abundance and (seemingly) source of acetate as well as the level of microbial

activity. The discussion did not dig into what might make this possible, to an extent that supports it as the favored mechanism. A second explanation for the identical acetate-water hydrogen isotope fractionation is that acetate and water are at isotopic equilibrium in the fluids, but with an EIE that differs significantly from the value calculated via DFT. This would contradict the main conclusion concerning acetate turnover. The authors identify acetate complexation with abundant (>1M) cations in the fluids as a possible matrix effect that could shift the EIE relative to the DFT-calculated value but argue against this possibility on the basis that the hydrophobic acetate methyl group would not participate in complexation with cations. While this seems reasonable in terms of a direct interaction between dissolved cations and methyl H, the proposed tautomerization mechanism for isotope exchange implies that the strength of the methyl C-H bond is affected by the adjacent carbonyl group. In turn, the carbonyl group should be strongly complexed in the highly saline fluids. Matrix cations could thus pull on the electrons in the C-H bond through the carbonyl group, and this might well affect the EIE. Perhaps the water side of the acetate-water equilibrium also stands to be affected by a high ionic strength? The text states that, "the speciation of acetate in salt mixtures and the resulting isotope effects are not constrained", and therefore that, "all available data suggest that acetate and water in Kidd Creek and Birchtree are in substantial isotopic disequilibrium". A different take is that the lack of experimental data concerning acetate speciation in salt mixtures leaves no basis for ruling out a simple explanation for the identical hydrogen isotope fraction in the two sites. To the extent that isotopic equilibrium in the acetate-water system cannot be ruled out, the study's main conclusion cannot be supported.

Additional experiments to determine the EIE in fluids with an ionic composition approximating that of Kidd Creek and Birchtree could serve to quantify any matrix effect, and thereby support or rule out the possibility that acetate and water are at isotopic equilibrium in the mine fluids. In the absence of such new data, it would be necessary to identify a concrete literature or theoretical basis on which to argue that a high ionic strength matrix effect could not cause the EIE to diverge from the DFT-calculated value by ~100%.

Discussion concerning mechanisms of acetate turnover and implications for microorganisms:

The balance of the paper is devoted to a discussion of the possible mechanisms of acetate turnover in crustal fluids, including the potential to provide energy to microorganisms in the case that they are responsible for acetate consumption. This discussion is only applicable to the specific case in which acetate turnover on <1Myr timescales is conclusively demonstrated, and as noted above, I do not think this has been accomplished yet. To the extent that it ultimately is, this section provides a balanced and well-presented discussion of the potential drivers and implications of acetate turnover in these systems.

Reviewer #2

(Remarks to the Author)

This manuscript describes a new approach to estimate acetate turnover in deep subsurface fluids. It is an interesting approach with potentially far-reaching implications. The use of different sites is good, but perhaps too few and too distinctly different (one being extremely old saline water and one being influenced by surface/near-surface water with plenty DOM and microbes). The influence of microbial taxa on the values could probably be expanded, and as such, more sites with different taxa would certainly aid in the interpretations of microbial influence. The analytics and methods seem adequate, although some more information is needed in places.

below are a few comments, both technical and general, on line-by-line basis:

Line 1: Title: Would be good to add "fluids" in the title.

Lines 33-35: Yes, but there are far more studied areas that should be mentioned here Fennoscandia, India etc, plenty of similar works there, albeit not as old and "isolated" as in Canadian shield and RSA.

Lines 33 and onwards: These sentences describe the really deep and highly saline brines detected in Canada and RSA. But there is also a zone in the upper 1-2 km that has highly mixed water of different salinities and origins. This is where the majority of the microbial communities thrive and where the bulk of the biomass of the deep biosphere is. It would be good to mention this here in the introduction, and focus more on it in the current paper.

Line 36: up to 325 g/L: this is probably not representative for the major part of the deep biosphere in this setting.

Line 59: Add "potentially" before other planets, as there are no known "deep biosphere location" elsewhere.

Line 60-75: Would be good also to dive deeper into how different microbial taxa may influence the isotopic values, is it different between different sites and taxa?

Line 97: Kidd Creek and Birchtree: The use of two sites is good, but they are really, really different. It would be good to have something intermediate between them to strengthen the model.

Fig1b: Add "acetate" on the x-and y-axes headings. Make the symbols different for the two areas, so they can be distinguished for the colorblind/grey scale print outs. Please indicate whether the error bars are 1 sigma or 2 sigma, and use preferably 2 sigma.

Line 159: >1 Gyr: this is not valid for Birchtree, is it?

Line 166-onwards: There is also biotic acetate formation in these systems. So that should be added as well and not just rates of acetate consumption. I do see that acetogenesis comes in the next chapter, but it should also be introduced already here.

Line 261: "external standard". Please refer to it as "reference material" instead and give the name of the material and what +/- the values given for the material are, and how these uncertainties have been propagated.

Suppl. materials: Fig S1: The Acetate $\delta^2\text{H}$ seems to be systematically higher in values than the standard. This should be taken into consideration in the propagation.

Fig S1: "Replicates of 1 mM and 2 mM acetate solutions yielded identical $\delta^{13}\text{C}$ and $\delta^2\text{H}$ values within analytical uncertainty, indicating that the extraction procedure is not fractionating." ->Fig S1 (top) show values for 2mM that are 1 permil

off, either high or low, but within 1 SD (barely). Using the term "identical" for these is a bit of a stretch. Instead it would be good to give some interpretation on why the 1 permil deviations exist, and how a 1 permil offset would influence the model, if propagated.

Version 1:

Reviewer comments:

Reviewer #1

(Remarks to the Author)

I thank the authors for performing a new set of DFT calculations in an attempt to quantify the influence of complexation on the equilibrium isotope effects. It is useful to see that, while such an effect does appear to exist, it is calculated to fall well short of the magnitude needed to explain the observed acetate-water fractionation. Although not definitive (as I believe the authors acknowledge), I believe this is sufficient to address the major concern expressed in my first review. Certainly, it is sufficient to warrant publication, such that readers can then make their own assessment.

Before finalizing the ms for publication, please double-check the calculations relating to Figure 3. When I tried to spot check these calculations, I find that, at an acetate consumption rate of 1 nM/yr, to achieve a catabolic power of $10^{-9.8}$ W/L (line 186) requires that acetate reacts with a Gibbs energy change of 5000 kJ/mol, about 100x higher than seems reasonable for this reaction. Perhaps I miscalculated, but it is worth double checking. If, instead, the values on line 186 are miscalculated, please confirm whether this is also the case for the results presented in Fig 3, and whether it influences any of the surrounding discussion.

Reviewer #2

(Remarks to the Author)

In general the authors have done a quite nice job in answering the questions/remarks that I had. In a way, they avoid to address the remarks by doing the suggested works and text changes asked for, and merely change/implement the more technical comments/changes, which is a bit of an easy path to take. Although, the explanations in the rebuttal are generally sufficient, I would like to see some more "disclaimers" to be added to the text for the most crucial caveats. One such thing is the use of data on the microbial taxa, which refer to MPN methods and enrichment cultures only. It is well known that enrichment cultures and MPN rarely reflect the diversity of subsurface taxa and their activity (metagenomes and transcriptomes from filtered fluids are recommended). I recommend that the microbial chapter (Constraining acetate sources and sinks in the subsurface Microbial metabolisms:) should clearly state what relevant knowledge that exist from microbiology studies at these sites (now it is focused on isotope values), what that tells us for the interpretations of influence of microbiology on the presented data, and also add that the lack of more rich and up to date omics data, i.e. (16S)/metagenomic/transcriptomic data actually makes it impossible to rule out acetogens being present at any of these sites (no matter what the enrichment culture/MPN bias might suggest). I understand that it can be extremely difficult to obtain metagenomic data from really deep sites like Kidd Creek, but for this site, some disclaimer that the full diversity is not known from the currently available and cited data.

REVIEWER COMMENTS

Reviewer #1 (Remarks to the Author):

The manuscript by Mueller et al reports the development of a hydrogen isotope exchange clock and its application to constrain the turnover time of acetate in Precambrian fracture fluids. The manuscript is clear and well written, and the subject matter and potential impact are suitable for Nature Communications. The development of the isotope exchange clock is supported by a strong experimental dataset and, as the authors suggest, this novel approach can be generalized across a range of compounds and timescales. The work therefore has potential to be an impactful contribution. However, in my opinion, the paper's primary conclusion concerning acetate turnover in Precambrian crustal fluids is not adequately supported. Specifically, as detailed below, that conclusion depends on establishing that acetate and water in the crustal fluids are in substantial hydrogen isotopic disequilibrium. The presently available data and accompanying discussion do not convincingly do so.

Isotope exchange clock:

The manuscript describes a series of time course deuterium labelling experiments that beautifully demonstrate exchange of protons between the methyl group of acetate and the water matrix, ultimately leading to isotopic equilibration between the two pools. The experiments further demonstrate a temperature dependence in the rate of isotopic exchange that adheres to a first order Arrhenius relationship over a temperature range from 60-200C. This relationship is used to place an upper limit on the rate of isotopic exchange at 25C. This is a novel approach that could have wide application in natural systems because, as the authors note, it is generalizable to a range of different compounds and equilibration timescales. This is an important and potentially impactful contribution.

We thank Reviewer 1 for their positive comments on the experimental results and the potential impact of the isotope exchange clock methods developed in this study. We believe that the development of this tool and the insights gained from it in the Kidd Creek system are the two most important contributions of this work.

Acetate turnover rates in natural systems:

The newly developed isotope exchange clock is used to place upper limits on the turnover time of acetate in fracture fluids from Kidd Creek and Birchtown mines, which differ in terms of microbial activity and the concentrations and carbon and hydrogen stable isotopic composition of acetate. It is argued that acetate is out of isotopic equilibrium with respect to the host fluids in both systems, such that the acetate turnover time must be shorter than the isotope equilibration timescale (< 1Myr). This is the major conclusion of the paper, and the subsequent discussion is focused on exploring the possible

mechanisms and implications of acetate turnover. This conclusion depends on demonstrating that a hydrogen isotopic disequilibrium exists between acetate and water. In my opinion, this is not convincingly demonstrated. The expected equilibrium isotope effect (EIE) at the mine fluid temperature of 25C is calculated via DFT, which was also used to calculate the EIE at temperatures ranging up to 200C. Experiments conducted at 200C demonstrate that acetate and MilliQ water equilibrate with an EIE that is within analytical error of the DFT-calculated value. This stands as a solid experimental demonstration that DFT accurately predicts the EIE at 200C in a pure water matrix. To demonstrate acetate-water disequilibrium in the mine fluids requires that the DFT calculations also accurately reflect the EIE at 25C in high ionic strength fluids.

The authors note that Kidd Creek and Birchtree exhibit nearly identical acetate-water hydrogen isotope fractionation, despite the considerable differences that led to their being chosen as contrasting sites for the present study. Two explanations are considered. The favored explanation is that the identical fractionation “likely reflects kinetic isotope effects that provide insight into acetate turnover mechanisms”. This certainly cannot be ruled out, but it would require that such effects yield identical fractionation in sites that differ in the abundance and (seemingly) source of acetate as well as the level of microbial activity. The discussion did not dig into what might make this possible, to an extent that supports it as the favored mechanism. A second explanation for the identical acetate-water hydrogen isotope fractionation is that acetate and water are at isotopic equilibrium in the fluids, but with an EIE that differs significantly from the value calculated via DFT. This would contradict the main conclusion concerning acetate turnover. The authors identify acetate complexation with abundant (>1M) cations in the fluids as a possible matrix effect that could shift the EIE relative to the DFT-calculated value but argue against this possibility on the basis that the hydrophobic acetate methyl group would not participate in complexation with cations. While this seems reasonable in terms of a direct interaction between dissolved cations and methyl H, the proposed tautomerization mechanism for isotope exchange implies that the strength of the methyl C-H bond is affected by the adjacent carbonyl group. In turn, the carbonyl group should be strongly complexed in the highly saline fluids. Matrix cations could thus pull on the electrons in the C-H bond through the carbonyl group, and this might well affect the EIE. Perhaps the water side of the acetate-water equilibrium also stands to be affected by a high ionic strength? The text states that, “the speciation of acetate in salt mixtures and the resulting isotope effects are not constrained”, and therefore that, “all available data suggest that acetate and water in Kidd Creek and Birchtree are in substantial isotopic disequilibrium”. A different take is that the lack of experimental data concerning acetate speciation in salt mixtures leaves no basis for ruling out a simple explanation for the identical hydrogen isotope fraction in the two sites. To the extent that isotopic equilibrium in the acetate-water system cannot be ruled out, the study’s main conclusion cannot be supported.

Additional experiments to determine the EIE in fluids with an ionic composition approximating that of Kidd Creek and Birchtree could serve to quantify any matrix effect, and thereby support or rule out the possibility that acetate and water are at isotopic

equilibrium in the mine fluids. In the absence of such new data, it would be necessary to identify a concrete literature or theoretical basis on which to argue that a high ionic strength matrix effect could not cause the EIE to diverge from the DFT-calculated value by ~100‰.

We thank the reviewer for holding our feet to the fire on this issue. We agree with the reviewer that the ionic strength of the experimental solution or environmental sample could impact the partition function ratio of both water and acetate. In response, we performed additional DFT calculations to determine the salt effect on acetate's hydrogen isotope composition and better approximate the equilibrium isotope effect (EIE) in a brine like that of Kidd Creek and Birchtree fluids.

Ca-Acetate Complex: It would require weeks or months of computational time to perform a single DFT calculation to explicitly model a unit cell of acetate solvated in water with intercalated ions. This calculation would need to be iterated and would have been prohibitively slow. Instead, we modelled the electronic structure of a calcium-acetate bidentate complex, which is known to form in concentrated CaCl₂ solutions, like Kidd Creek and Birchtree fluids (see Table S1). The Ca-acetate complex is the most likely to form in the samples as Ca²⁺ is >3 M in the samples. Taking this approach also directly addresses the reviewers concerns that “*matrix cations could thus pull on the electrons in the C-H bond through the carbonyl group, and this might well affect the EIE.*” A previous publication from Noval et al. (2018) determined the molecular geometry of the Ca-acetate complex (i.e., bond lengths and angles). Using their geometry as the initial condition on a DFT calculation, we optimized the molecular structure *in silico* and then calculated the vibrational frequencies of the complex with and without single deuterium (D) substitutions on the methyl-site. The same level of theory and linear regression from Wang et al. (2009) were used as in the main text. These calculations gave us the reduced partition function ratio of the calcium-acetate complex (β_{complex}).

Water: We also corrected the water partition function ratio (β_{water}) for a salt effect that has been determined empirically in CaCl₂ brines, as suggested by the reviewer. Horita et al. (1992) found that β_{water} increases by 15‰ in CaCl₂ brines relative to pure water at room temperature. We applied this salt effect to our calculated β_{water} value to derive the partition of the brine (β_{brine}). We then calculated the EIE across a series of temperatures, expressing these values as ${}^2\alpha_{\text{complex/brine}}$ for the Ca-acetate complex and ${}^2\alpha_{\text{acetate/brine}}$ for the free acetate molecule using the following equations:

$$\beta_{\text{brine}} = 1.015 \times \beta_{\text{water}}$$
$${}^2\alpha_{\text{complex/brine}} = \frac{\beta_{\text{complex}}}{\beta_{\text{brine}}}$$

$${}^2\epsilon_{\text{complex/brine}} = ({}^2\alpha_{\text{complex/brine}} - 1) * 1000$$

As shown in Figure R1, ${}^2\epsilon_{\text{complex/brine}}$ is different from ${}^2\epsilon_{\text{acetate/brine}}$ and this offset expands at lower temperatures. At 25°C, ${}^2\epsilon_{\text{complex/brine}}$ is -166‰ while ${}^2\epsilon_{\text{acetate/brine}}$ is -204‰. Thus, the reviewer is correct that complexation affects the EIE between acetate and water. We hypothesize that the offset between β_{complex} and β_{acetate} is due to a decrease in the *in silico* optimized C-O-C bond angle in the Ca-acetate complex (121.3°) compared to the free acetate molecule (127.2°). Nevertheless, the measured difference between $\delta^2\text{H}_{\text{acetate}}$ and $\delta^2\text{H}_{\text{water}}$ in Kidd Creek is -98‰ to -110‰ and still greatly exceeds the newly predicted equilibrium. Even incorporating this complexation then, acetate in the fracture fluid samples are >50‰ more D-enriched than the EIE would predict (Figure R1). Furthermore, the ${}^2\epsilon_{\text{complex/brine}}$ represents a conservative estimate which assumes that every acetate molecule is complexed to calcium.

Figure R1: The calculated EIE between acetate and water (dashed line) and Ca-acetate complex and water (solid line) across a range of temperatures. Samples from Kidd Creek (blue dots) and Birchtree (red dots) are offset from the EIE lines, indicating that they are in isotopic disequilibrium regardless of whether acetate is complexed. The shaded region represents all possible EIE values due to mixing between the free acetate and Ca-acetate complex pools.

Ideally, the EIE could be tested empirically as the reviewer also suggests. Unfortunately, due to the slow kinetics of hydrogen isotope exchange, full equilibration of acetate with water at low temperatures (<200C) would require months of experimental time which we do not have the resources to execute. At 25C, it would require thousands of years. We have already determined ${}^2\epsilon_{\text{acetate/water}}$ at 200C and this could be repeated with brines, but the results would likely be inconclusive since differences between ${}^2\epsilon_{\text{acetate/water}}$ and ${}^2\epsilon_{\text{complex/brine}}$ values are small at higher temperatures (Figure R1). These experiments would realistically require >2 months to prepare, execute, and analyze, and they would not constrain the salt effect at lower temperatures.

In summary, these new insights demonstrate that acetate complexation with cations does somewhat impact the EIE with water, but importantly does not explain the observed difference between the hydrogen isotope compositions of water and acetate in the subsurface fluids. We thank the reviewer for this suggestion as the additional calculations further strengthen the original conclusions of the study, specifically that acetate and water are in isotopic disequilibrium. The primary conclusion of the study – that acetate is being actively produced and consumed in the subsurface – is unaffected by these considerations.

We have expanded on the section of the Results describing complexation to succinctly describe these findings. Figure 2B has also been adjusted to show EIE for the complex, free acetate, and a shaded region indicating a mixture of these two end-members. We have also added a description of our DFT calculations for the Ca-acetate complex, including its optimized geometry in the Supplemental Information. We have added a new coauthor (A Meshoulam) who helped us with these rather difficult calculations. We thank the reviewer for insightfully drawing our attention to this issue. Our findings and interpretations are improved because of their comments.

Discussion concerning mechanisms of acetate turnover and implications for microorganisms:

The balance of the paper is devoted to a discussion of the possible mechanisms of acetate turnover in crustal fluids, including the potential to provide energy to microorganisms in the case that they are responsible for acetate consumption. This discussion is only applicable to the specific case in which acetate turnover on <1Myr timescales is conclusively demonstrated, and as noted above, I do not think this has been accomplished yet. To the extent that it ultimately is, this section provides a balanced and well-presented discussion of the potential drivers and implications of acetate turnover in these systems.

Thank you for the positive remarks on the content of the Discussion section. We believe that with the further computational models provided in the revised manuscript we have sufficiently demonstrated that acetate and the fracture fluid waters are in disequilibrium and that acetate is being actively turned over in the fluids. Therefore, the remainder of the discussion remains the same, other than changes made in response to Reviewer 2's comments.

Reviewer #2 (Remarks to the Author):

This manuscript describes a new approach to estimate acetate turnover in deep subsurface fluids. It is an interesting approach with potentially far-reaching implications. The use of different sites is good, but perhaps too few and too distinctly different (one being extremely old saline water and one being influenced by surface/near-surface water with plenty DOM and microbes). The influence of microbial taxa on the values could

probably expanded, and as such, more sites with different taxa would certainly aid in the interpretations of microbial influence. The analytics and methods seem adequate, although some more information is needed in places.

below are a few comments, both technical and general, on line-by-line basis:

Line 1: Title: Would be good to add "fluids" in the title.

We have amended the title to include the term "fluids" for clarification.

Lines 33-35: Yes, but there are far more studied areas that should be mentioned here Fennoscandia, India etc, plenty of similar works there, albeit not as old and "isolated" as in Canadian shield and RSA.

Lines 33 and onwards: These sentences describe the really deep and highly saline brines detected in Canada and RSA. But there is also a zone in the upper 1-2 km that has highly mixed water of different salinities and origins. This is where the majority of the microbial communities thrive and where the bulk of the biomass of the deep biosphere is. It would be good to mention this here in the introduction, and focus more on it in the current paper.

We have added this to the introduction.

Line 36: up to 325 g/L: this is probably not representative for the major part of the deep biosphere in this setting.

Thank you for the preceding three comments. All three have been addressed in the main text Introduction by specifying that the manuscript highlights primarily the deeper (>1-2km) fracture fluids that are primarily old and isolated (Kidd Creek) or mixed with paleometeoric waters (Birchtree) but not than those of the near surface which mix readily with modern surface waters.

Line 59: Add "potentially" before other planets, as there are no known "deep biosphere location" elsewhere.

We have made this change in the main text.

Line 60-75: Would be good also to dive deeper into how different microbial taxa may influence the isotopic values, is it different between different sites and taxa?

We thank the reviewer for raising an important point. The influence of microbial taxonomy on isotopic values of acetate are not well understood. It has been shown that the mechanism of acetate activation during its consumption can affect the carbon isotope fractionation on acetate by acetoclastic methanogenesis and sulfate reduction. However, acetate activation strategy does not necessarily align with taxonomy. Furthermore, there are only limited studies of microbial communities in fracture fluids of Birchtree or Kidd Creek to contextualize such a discussion. Notably, Birchtree has a higher cell density and hosts a more diverse set of microbial metabolisms than Kidd Creek fluids based on most probably number (MPN) experiments. MPN experiments are limited in that they only consider culturable microorganisms, a small fraction of the natural population. However, it is clear from MPN analyses of Birchtree fluids that fermenters and organoclastic sulfate reducers are both present. The same cannot be said for Kidd Creek fluids where only

hydrogenotrophic and alkane-oxidizing sulfate reducers could be enriched. We have added several sentences at the end of the introduction to inform the reader of these differences and of the lack of taxonomic information in both mines. We believe this information gives the reader helpful context for the subsequent discussions of acetate sources and sinks in the Discussion.

Line 97: Kidd Creek and Birchtree: The use of two sites is good, but they are really, really different. It would be good to have something intermediate between them to strengthen the model.

The reviewer brings up an important caveat of our work. We recognize that more environmental data would strengthen the discussion section regarding the potential biogeochemical sources and sinks of acetate in the subsurface. However, retrieving samples from such locations is difficult and would require further sample preparation and instrumental analysis, which we do not presently have the resources or time to perform. We contend that further contextual evidence provided by other environmental data would not impact the substantive conclusion of our study – acetate is being turned over in extremely isolated locations within Precambrian continental crust fluids. If acetate were in isotopic equilibrium at other sites, it would certainly be an interesting finding and would warrant further characterization of those sites' biogeochemistry (e.g. microbial community composition, geochemistry, and radiolytic output) to compare the sites and determine the influences on acetate turnover. However, this is outside the scope of our study and would not change the conclusions of our results. We present and describe a novel method to infer acetate turnover, one that can be employed in such future investigations, even in the most isolated systems like Kidd Creek. We believe the environmental data presented in the revised manuscript is sufficient to make the claims in the title and discussion that acetate is actively turned over in the subsurface.

Fig1b: Add "acetate" on the x-and y-axis headings. Make the symbols different for the two areas, so they can be distinguished for the colorblind/grey scale print outs. Please indicate whether the error bars are 1 sigma or 2 sigma, and use preferably 2 sigma.

This edit has been made in the main text.

Line 159: >1 Gyr: this is not valid for Birchtree, is it?

We have specified that we are discussing Kidd Creek in that sentence.

Line 166-onwards: There is also biotic acetate formation in these systems. So that should be added as well and not just rates of acetate consumption. I do see that acetogenesis comes in the next chapter, but it should also be introduced already here.

Thank you for this comment. We agree that it is important to recognize both sources and sinks of acetate in the main text. This particular section only discusses the potential for microbial consumption. We felt this deserved to be at the top of the discussion because it is a calculation that can be directly ascertained from the turnover rates estimated with our methods. Whereas, the exact source of acetate (whether microbial or abiotic or both) can

only be constrained with circumstantial evidence. We felt it was prudent to discuss the implications of our findings that are backed by firm quantitative arguments before those that call for more speculation.

We discussed the evidence supporting and denying microbial/abiotic sources and draw a putative conclusion in a subsequent section of the Discussion. We should not assume that there is biotic acetate formation in these systems, particularly in the Kidd Creek formation where no culturable fermenters or acetogens have been found. This does not eliminate the possibility that fermenters and acetogens are active in the KC fracture fluids, as we note in the main text. However, a biotic origin of acetate does not easily explain the isotopic data when contextualized against other environmental measurements of acetate $\delta^{13}\text{C}$, including at the Birchtree mine where fermenters are demonstrably present. As discussed in the main text, the exact sources of acetate cannot be confirmed from the available data, only its minimum turn over rates.

Line 261: "external standard". Please refer to it as "reference material" instead and give the name of the material and what +/- the values given for the material are, and how these uncertainties have been propagated.

We thank the reviewer for paying close attention to the analytical methods at the foundation of our conclusions. We have altered the text to say "working standard". We are hesitant to call these "reference materials" as they have not been characterized through a ring-test across different laboratories nor have they been blessed by either IAEA, NIST, or USGS as are most "reference materials". To avoid confusion with the broader isotope geochemistry community, we refrain from using the term "reference material".

The uncertainties on the true value of these working standards have been added into the text. In terms of propagation, the standard deviations in the paper are calculated from triplicate measurements of sample-standard comparisons that each yield a 'delta' value ($\delta^2\text{H}$ and $\delta^{13}\text{C}$). The uncertainty on the working standard value is ignored in this calculation. This approach is denoted the "external reproducibility" and is a more conservative assessment of the total error on the analytical method. If instead the internal reproducibility of each replicate measurement and the uncertainty on the working standard are added in quadrature to propagate their uncertainties, the total error is smaller than the external reproducibility. This is because the internal reproducibility ($\sim 0.4\text{‰}$) is quite low and the uncertainty on the working standard is even lower (0.1‰). As a result, we chose the conservative estimate for the total uncertainty, which is the external reproducibility. This is discussed in the original manuscript describing the ESI-Orbitrap method for measuring acetate's isotope composition. We have added a brief statement regarding uncertainty calculations to the Methods section to address this confusion.

Suppl. materials: Fig S1: The Acetate $\delta^2\text{H}$ seems to be systematically higher in values than the standard. This should be taken into consideration in the propagation.

Fig S1: "Replicates of 1 mM and 2 mM acetate solutions yielded identical $\delta^{13}\text{C}$ and $\delta^2\text{H}$ values within analytical uncertainty, indicating that the extraction procedure is not

fractionating." ->Fig S1(top) show values for 2mM that are 1 permil off, either high or low, but within 1 SD (barely). Using the term "identical" for these is a bit of a stretch. Instead it would be good to give some interpretation on why the 1 permil deviations exist, and how a 1 permil offset would influence the model, if propagated.

Every measurement of both $\delta^{13}\text{C}$ and $\delta^2\text{H}$ are within 2SD of the known value. Minor offsets (<1‰) between expected and measured values are within normal operating conditions for ESI-Orbitrap measurements of compound-specific $\delta^{13}\text{C}$ on acetate, as described in the original method development study (Mueller et al. 2022). As such, the accuracy of extracted samples are similar to the reported accuracy on pure solutions of sodium acetate standards. Thus, the extraction procedure did not produce any isotopic fractionations during measurements beyond those that are already inherent to the method itself. The uncertainties on acetate $\delta^{13}\text{C}$ measurements are already over 0.5‰. Increasing these error bars by 1‰ does not change the interpretation that Kidd Creek acetate is highly ^{13}C -enriched. These values are not used in the quantitative model of turnover times, which only rely on the $\delta^2\text{H}$ values.

REVIEWER COMMENTS

Reviewer #1 (Remarks to the Author):

I thank the authors for performing a new set of DFT calculations in an attempt to quantify the influence of complexation on the equilibrium isotope effects. It is useful to see that, while such an effect does appear to exist, it is calculated to fall well short of the magnitude needed to explain the observed acetate-water fractionation. Although not definitive (as I believe the authors acknowledge), I believe this is sufficient to address the major concern expressed in my first review. Certainly, it is sufficient to warrant publication, such that readers can then make their own assessment.

Before finalizing the ms for publication, please double-check the calculations relating to Figure 3. When I tried to spot check these calculations, I find that, at an acetate consumption rate of 1 nM/yr, to achieve a catabolic power of $10^{-9.8}$ W/L (line 186) requires that acetate reacts with a Gibbs energy change of 5000 kJ/mol, about 100x higher than seems reasonable for this reaction. Perhaps I miscalculated, but it is worth double checking. If, instead, the values on line 186 are miscalculated, please confirm whether this is also the case for the results presented in Fig 3, and whether it influences any of the surrounding discussion.

Response: Thank you to the reviewer for catching this important mistake in the main text (but happily affecting only the words in the text and not Figure 3). These are indeed typos from earlier drafts of the manuscript. We have changed these to $10^{-11.5}$ and 10^{-12} W/L for sulfate reduction and methanogenesis, respectively. These correspond to Gibbs energy of -98 kJ/mol and -30 kJ/mol for these two metabolisms. We have thoroughly checked the calculations used to make Figure 3 and they are all accurate.

Reviewer #2 (Remarks to the Author):

In general the authors have done a quite nice job in answering the questions/remarks that I had. In a way, they avoid to address the remarks by doing the suggested works and text changes asked for, and merely change/implement the more technical comments/changes, which is a bit of an easy path to take. Although, the explanations in the rebuttal are generally sufficient, I would like to see some more "disclaimers" to be added to the text for the most crucial caveats. One such thing is the use of data on the microbial taxa, which refer to MPN methods and enrichment cultures only. It is well known that enrichment cultures and MPN rarely reflect the diversity of subsurface taxa and their activity (metagenomes and transcriptomes from filtered fluids are recommended). I recommend that the microbial chapter (Constraining acetate sources and sinks in the subsurface Microbial metabolisms:) should clearly state what relevant knowledge that exist from microbiology studies at these sites (now it is focused on isotope values), what that tells us for the interpretations of influence of microbiology on the presented data, and also add that the lack of more rich and up to date omics data, i.e. (16S)/metagenomic/transcriptomic data actually makes it impossible to rule out

acetogens being present at any of these sites (no matter what the enrichment culture/MPN bias might suggest). I understand that it can be extremely difficult to obtain metagenomic data from really deep sites like Kidd Creek, but for this site, some disclaimer that the full diversity is not known from the currently available and cited data.

Response: We thank the reviewer for this and to address all of the relevant microbial and genetic data that is available at these sites more fully, we have added a brief description of the two studies published to date that have relevant information on the Kidd Creek fracture fluids to the "Microbial sources and sinks of acetate" section of the manuscript. We believe this enables the reader to make their own judgments regarding the potential sources and sinks of acetate in the samples as we continue to search for more definitive evidence.